# Effectiveness of Virtual Reality and Feedback to Improve Gait and Balance in Patients with Diabetic Peripheral Neuropathies: Systematic Review and Meta-Analysis

**DOI:** 10.3390/healthcare11233037

**Published:** 2023-11-24

**Authors:** Laura Alonso-Enríquez, Laura Gómez-Cuaresma, Maxime Billot, Maria Isabel Garcia-Bernal, Maria Luisa Benitez-Lugo, María Jesús Casuso-Holgado, Carlos Luque-Moreno

**Affiliations:** 1Faculty of Nursing and Physiotherapy, University of Cadiz, 11009 Cadiz, Spain; laura.alonsoenriquez@alum.uca.es; 2IntegraSalud Centro de Rehabilitación Funcional, El Puerto de Santa María, 11500 Cadiz, Spain; 3Hôpital Du Pays De L’Autan, 81100 Castres, France; laura.gomezcuaresma@alum.uca.es; 4PRISMATICS Lab (Predictive Research in Spine/Neuromodulation Management and Thoracic Innovation/Cardiac Surgery), Poitiers University Hospital, 86021 Poitiers, France; maxime.billot@chu-poitiers.fr; 5Instituto de Biomedicina de Sevilla (IBIS), 41013 Seville, Spain; ibernal@us.es (M.I.G.-B.); mcasuso@us.es (M.J.C.-H.); 6Department of Physiotherapy, University of Seville, 41009 Seville, Spain; marisabeni@us.es

**Keywords:** Diabetic Neuropathies, Virtual Reality, Feedback, Sensory, Gait Disorders, Neurologic, Postural Balance

## Abstract

Diabetic peripheral neuropathy (DPN) is the primary complication in patients with diabetes mellitus, characterized by loss of sensation and function in the lower limbs. Virtual reality (VR) and/or sensory feedback (FB) therapy has shown positive effects in other neurologic conditions such as stroke. However, consensus regarding their effectiveness in the DPN population is lacking. This study aims to analyze existing scientific evidence about the effects of VR and/or FB on improving gait and balance and reducing the risk of falls in patients with DPN (pwDPN). A thorough search was conducted in scientific databases including PubMed, Scopus, and EMBASE, up until November 2023. CMSQ, the PEDro scale, and the Cochrane Collaboration’s tool were used to assess the methodological quality and risk of bias of the studies. A total of 10 studies were selected for qualitative analysis, with three contributing information to the meta-analysis. The combined results suggest a positive trend in favor of VR and FB rehabilitation; however, significant differences were not observed in balance (SMD = −0.81, 95% CI = −1.90, 0.29; *p* = 0.15; I^2^ = 86%) or gait speed improvements (MD = −1.05, 95% CI = −2.96, 0.85; *p* = 0.28; I^2^ = 89%). Therefore, further randomized controlled studies are still needed to achieve stronger conclusions regarding the benefits of VR and/or FB in pwDPN.

## 1. Introduction

Recently, diabetes mellitus (DM) has emerged as a significant disease with an increasing prevalence rate. It is estimated that 462 million individuals are affected by type 2 diabetes, accounting for approximately 6.3% of the world’s population, and this number has continued rising, reaching 6059 cases per 100,000 people a year [1]. Diabetic Peripheral Neuropathy (DPN) is the main neuromuscular complication and the most prevalent chronic consequence, affecting about 73% of the DM population [2]. DPN is characterized by progressive degeneration of nerve fibers, primarily in the lower limbs, leading to decreased signal conduction velocity [3] and resulting in symptoms such as hypoesthesia, poor proprioception, impaired reflexes [4], reduced strength [5] in lower limbs, and decreased step length and functional gait speed [6]. Furthermore, the causes of proprioception alteration do not fully explain the variability in pinch proprioception in the DPN population, indicating a greater role of muscle spindles [7]. People with DPN (pwDPN) exhibit a deficiency in both static (the ability to maintain an upright stance while managing center of mass (CoM) sway [8]) and dynamic balance (the ability to maintain balance during movement) [9,10]. Specifically, there is a notable decline in static balance, as patients appear to acquire anticipatory postural strategies during movement [11]. These strategies pertain to proactive balance aimed at preventing disturbances, in contrast with reactive balance which responds to perturbations [12]. While balance is regulated by proprioception and exteroception, and visual and vestibular systems, [8,9,13], their impairment in pwDPN compromises balance. Thereby, pwDPN heavily rely on visual input for standing and walking, increasing the likelihood of falls in closed-eye conditions [14,15,16]. This elevates the risk of falls (causing damage more than 15 times greater than in subjects without DPN [17]), especially in the medial–lateral plane due to increased CoM sway [18,19,20]. In pwDPN, gender and obesity seem to have more influence on balance control than age, as men exhibit greater sway than women, particularly without visual input, and obese subjects experience an increase in sway without vision [21]. On a global scale, alteration of visual cues due to diabetic retinopathy [22,23] and dysfunction in the vestibular system [24] contribute to balance deficits and an increased risk of falls in pwDPN.

Instability and alteration in gait parameters constitute the primary causes of falls—one of the most serious complications experienced by pwDPN [9,14,15,19]. Even in the initial stages of the DPN disease, altered spatio-temporal gait parameters have been observed, both with eyes closed and open, likely due to an alteration of knee extensor activation, compensated for by ankle and hip strategies [25]. Furthermore, the long-term lack of sensory input leads to other severe complications such as ulceration, amputations, diabetic foot, and an increased risk of falls [8,26], reducing quality of life [20,27] and increasing healthcare costs.

The primary goals of physiotherapy for pwDPN include enhancing balance, proprioception, and gait quality. This approach aims to limit mortality and morbidity by preventing foot deformities and improving the distribution of plantar load, which is related to high-pressure areas during walking, thereby decreasing the risk of ulceration [28,29,30]. To achieve these goals, conventional physiotherapy has focused on specific balance and gait programs [8] as well as tailored orthosis [31]. Various interventions, such as electrotherapy and magnetotherapy [32], exercise therapy [6], and programs specifically targeting balance and gait [8], have demonstrated a high level of evidence. Other complementary approaches, such as acupuncture [33] and kinesiotaping [34], have shown relative evidence, while offloading devices do not appear to improve static and dynamic balance [35].

Recently, innovative technology like virtual reality (VR) and feedback (FB) have arisen and are widely used as complementary therapies for gait and balance disorder rehabilitation in various neurological conditions [36,37,38] and, to a lesser extent, in DPN. VR-based treatment includes computerized programs specifically designed to recreate real environments [39], fostering a secure clinical space supported by medical staff while improving motivation and adherence [40]. There are three main types of VR based on interaction levels: non-immersive [41], featuring limited interaction and environments generated by small devices like screens or joysticks; semi-immersive systems, using larger devices or setups like Kinect [42,43]; and immersive environments, creating controlled and highly interactive settings with a greater sense of presence and stimulation. VR integrates the concept of FB, harnessing the ability to voluntarily control and modify certain bodily functions or biological processes with additional input [44]. FB is categorized based on the input: visual, auditory, or tactile/haptic [44,45]. FB appears crucial when shaping new movement patterns or balance schemas in patients experiencing sensory loss. Relevant learning occurs during both the execution of the movement or the goal of the exercise [46], providing crucial information and enhancing patients’ motor learning.

The literature indicates that VR has shown clinical benefits in gait and balance rehabilitation among individuals with neurological disorders, such as stroke [37], Parkinson’s disease [36], and cerebral palsy [38], and even in the recovery from orthopedic ankle injuries [47]. Despite extensive studies on pharmacological [48] and non-pharmacological therapies [31,32,33,34] for pwDPN rehabilitation, a systematic review focusing on the use of VR in pwDPN is lacking. Peripheral neurological involvement alters the body schema at the central level, suggesting potential benefits for pwDPN through tailored VR programs. The aim of the current study is to comprehensively review and synthesize existing evidence regarding the effects of both VR and FB on improving gait, balance, and risk of falls in pwDPN.

## 2. Materials and Methods

The requirements of “Preferred Reporting Items for Systematic Review and Meta-Analysis (PRISMA) Model” [49] have been followed in this systematic review. The protocol of this systematic review has been registered in the PROSPERO database (CRD42021255708).

### 2.1. Search Strategy

Up until November 2023, an exhaustive search of studies was conducted in the databases: Medline/PubMed, Scopus, and EMBASE. The search strategies in the different databases are shown in Table 1.

### 2.2. Eligibility Criteria

The studies selected for this systematic review were chosen after applying the PICO inclusion criteria [50]: (P) in patients with peripheral neuropathy in lower limb due to diabetes, (I) is effective virtual reality and/or feedback (C) compared to other intervention, non-intervention, or without comparison (O) to improve gait, balance and/or risk of falls?(s) Study Design: Randomized control trials, non-randomized control trials, prospective cohort studies, case series, crossover trials, pilot studies, and clinical trials. 

The “human” search filter was used and the search was limited to studies in English and Spanish.

The exclusion criteria were: studies did not carry out an intervention, were not focused on gait and/or balance, or were focused solely on the evaluation, treatment, and/or monitoring of plantar pressures predisposing to ulceration.

### 2.3. Assessment of the Methodological Quality and Risk of Bias

The evaluation of methodological quality was carried out through the Physiotherapy Evidence Database (PEDro) scale [51] in the studies for which it was possible to implement this scale (RCTs), and the Checklist for Measuring Study Quality (CMSQ) [40] for all the studies included (to obtain a broader vision of all types of studies such as case series, etc.).

The PEDro scale [51] was developed to identify studies that tend to be valid internally and to redirect clinical decision-making. It consists of 11 criteria, granting a point for each criterion met. This scale shows the degree of methodological quality of a given study, with a score ranging from 0 to 10. Those with a score of 9–10 points on the PEDro scale are considered to have excellent methodological quality; between 6–8 points, good methodological quality; between 4–5 points, fair methodological quality; and fewer than 4 points, poor methodological quality [52]. 

The CMSQ [53] consists of 27 “yes” or “no” questions organized into five sections: (1) study quality; (2) external validity; (3) study bias; (4) confusion and selection bias; and (5) study power. It presents a scoring system for individual sections and an overall final score.

The Cochrane Collaboration’s tool [54] was used to analyze the risk of bias in the RCTs. To evaluate, it includes six domains and uses three terms: “low risk”, “high risk”, and “unclear risk” and is represented by a table and a chart. 

Two authors (L.A.-E. and L.G.-C.) carried out quality assessment of the studies by implementing the different scales, and an additional reviewer (C.L.-M.) was considered for consensus when needed.

### 2.4. Selection Process and Data Extraction

The selection process consisted of several stages. A first pre-search was performed in the main database used, PubMed, to obtain the most appropriate search terms for search strategies. A definitive search was subsequently implemented in all the databases with their different strategies, as shown in Table 1. Once complete, duplicates were removed and a detailed title and abstract began to be read, identifying studies of high relevance. After that, we verified the studies that fell within the inclusion criteria established for this review by reading the full text. Finally, a last bibliographic search was done manually, and two more studies were identified. 

Two authors (L.A.-E. and L.G.-C.) carried out the screening process, and an additional reviewer (C.L.-M.) was considered for consensus when needed. The data extracted from the selected studies were author, year of publication, patient characteristics (total number of participants and by groups, average age, sex), disease evaluation, intervention details such as type VR or BF, sessions/repetitions, follow-up/reviews, as well as outcomes, measurement instruments, and finally results.

### 2.5. Data Synthesis and Statistical Analysis

The findings were described narratively and where possible, study results were pooled and meta-analysis was conducted based on the same underlying outcomes (gait parameters, balance). Mean differences or standard mean differences (SMD) were calculated with 95% confidence intervals (CI). Fixed or random effects models were used according to the degree of heterogeneity (I^2^ coefficient ≤ 50%), assuming a 95% confidence interval (CI) for all analyses. Review Manager software (RevMan v.5.4.1, The Cochrane Collaboration, 2020) was used to summarize the effects and construct the forest plots. EPIDAT software version 3.1. was used to check the risk of publication bias of the studies included in our meta-analysis. Whenever possible, Begg and Egger values were obtained (*p* < 0.05).

## 3. Results

A total of 137 studies were identified (the sum of the results found in the aforementioned databases), from which only 10 studies were finally selected for this systematic review, after applying the eligibility criteria (Figure 1). A list was made to show the excluded studies and the reasons for exclusion, which is found as Appendix A.

### 3.1. Methodological Quality and Risk of Bias of Included Studies

Table 2 shows the score on the PEDro scale [51] resulting from the two RTCs [10,17] selected for this review. The overall score is 7 on average, illustrating the good methodological quality of the selected studies.

Regarding the risk of bias of RCTs, all studies presented the same risk of bias. With respect to the evaluation of the different types of biases according to the selected studies, the items that obtained the highest biases were the blinding of participants and staff (performance bias), and the blinding of outcome assessments (detection bias). The results are shown in Figure 2 and Figure 3.

Finally, the results obtained through the CMSQ [53] were highly variable, obtaining an average score of 16 and a total range of 5 to 22. In general, the best scores were obtained for the RCTs [10,17]. The results are shown in Table 3.

### 3.2. Synthesis of Results

Table 4 shows the most relevant characteristics of the included studies. As is shown, all studies reported a positive effect of VR and FB interventions on balance and gait in pwPDN. When compared with other interventions, pooled data demonstrated a tendency in favor of VR and FB, but differences between groups were not statistically significant for balance (SMD = −0.81, 95% CI = −1.90, 0.29; *p* = 0.15; I^2^ = 86%) or gait speed (MD = −1.05, 95% CI = −2.96, 0.85; *p* = 0.28; I^2^ = 89%) (Figure 4 and Figure 5).

### 3.3. Participant and Intervention Characteristics

The total number of participants was 251 patients with DPN plus 71 healthy participants: 30 in the study by Mohieldin et al. [58] and 40 in the study by Stolarczyk et al. [56]. Neither of these two studies specified the participants’ gender, nor did Grewal et al. [59] in 2013, but it was concluded that there were 137 women and 105 men, with ages between 49.7 and 80.8 years. All of the participants were elderly adults and only in Mohieldin et al. [58] was the average age under 50 years.

The two publications reported by Salsabili et al. [8,49] seem to be part of the same study, since the number of participants, average age, and intervention dose were the same, distributing analysis of the outcome measures between the two publications.

#### 3.3.1. Classification of Participants According to Peripheral Neuropathy

Regarding the way to measure the severity or even the presence of DPN, different instruments and evaluative tests were used. 

Grewal et al. [17,59] chose the Vibration Perception Threshold (VPT), with a perception threshold >25 V or, if it was accessible, using the 10-g monofilament test on the heads of the first, third, and fifth metatarsal. In the studies by Salsabili et al. [9,60], the presence of DPN was evaluated by measuring the conduction velocity of the peroneal and tibial motor nerves and the sural sensory nerve. Once the presence of DPN is confirmed, its severity is tested with the Valk test, a 10-item instrument that assesses polyneuropathy severity according to the frequency of symptom appearance. Taveggia et al. [10] also used a monofilament test, in addition to the Diabetic Neuropathy Index criteria [61], an index used for over 20 years. Moreover, Mohieldin et al. [58] followed the criteria established by Dyck et al. [62] and divided all the participants into four categories: without DPN, asymptomatic DPN, moderate DPN, or severe DPN. The rest of the studies did not measure DPN severity. 

#### 3.3.2. Training Sessions

Training session duration ranged from 10 to 60 min, including breaks between tasks. Interventions were carried out using videogames created specifically for rehabilitation: MatLab [17,59], OGRE [27], or XavixPORT [57] and/or dynamometric platforms, Biodex Stability System [9,10,56,60], and SMART Balance Master [58]. Only one of the studies used wearable vibration [55] on key muscles, while another used an ax treadmill with specific software (Gait trainer 3 Biodex) and an electromechanical dynamometer (Biodex System 4) [10]. 

Grewal et al. [17,59] included a series of point-to-point protocols where the patient had to coordinate the hips and upper limbs to get from one point to another while standing, receiving visual FB of CoM Sway and obtaining positive auditory FB for good performance. The 2015 [17] study included another point-to-point protocol with obstacles. 

Salsabili et al. [9,60] proposed complete training using the Biodex Stability System platform (connected to the specific software Biodex 945-302 (Biodex Medical Systems Inc., Shirley, NY, USA) consisting of two limits of stability with parallel feet, three weight shifts, first with parallel feet, and then changing the advanced one, and the one standing stable, with visual FB of the patient’s CoM Sway.

This platform was also used by Stolarczyk et al. [56], including training protocolized by the procedure itself. Game difficulty was chosen individually according to patient characteristics and increased as they improved their results, but there was no clear explanation of the games themselves. 

Taveggia et al. [10], used a treadmill, an electromechanical dynamometer, and a Biodex dynamometric platform, with 20 min of gait, 20 min of soft strengthening, and 20 min of balance protocol, with specific attention paid to haptic FB, which was used in only a minority of studies. The control group performed training of similar duration with manual strengthening and stretching exercises, in addition to 20 min of walking and standard static and dynamic balance protocols. 

Mohieldin et al. [58] used dynamic posturography as the evaluation instrument in both the intervention and control group, but training was in the intervention group only, which received visual FB of their CoM, while the control group received verbal and/or haptic input. In this study, the doses were not similar: while both groups trained for three months, the intervention group had two training sessions a week and the control group had three. 

Using the OGRE game, Carroll & Galles [27] developed three specific interactive videogames for their patients. The first was to collect coins (visual FB), the second to steer a boat and shoot lasers (combining moves from MMII and MMSS), and the third to dodge obstacles with a plane. The patient interacted with the games through the Microsoft Kinect camera. 

Hung et al. [57] similarly proposed training based on the visual and auditory FB that the games created specifically for this purpose through the XavixPORT console. It was divided into four exercises: knee raises following the reference on the screen, stepping on hamsters, following the rhythm (first fast and then slow), and repeating the knee raises. In this study, the intervention was cross-sectional, which means that group A trained first and group B served as the control, while after the intervention time, group B trained and group A served as the control. 

Chandrashekhar et al. [55] were the only authors to use a localized wearable (Myovolt) in the tibialis anterior, quadriceps, and muscle belly of the gastrocnemius and soleus at different speed frequencies to provide different haptic FB to the participants in their study: group 1 received an intermittent 120 Hz vibration; group 2, a 35–120 Hz sinusoidal, and group 3, a 120 Hz continuous vibration.

### 3.4. Study Groups Included in the Statistical Analysis

It was possible to pool data from only three studies [9,21,47] and two outcomes: balance and gait speed (Figure 4 and Figure 5).

In all the studies, to a greater or lesser extent, balance improved in some of its components: postural control or static balance assessed by stabilometric platform [9] or the Unipedal Stance Test [47] and functional or dynamic balance, assessed by the Berg Balance Scale [47] and Tinetti Scale [9]. Pooled data from these three clinical trials [9,21,47] showed better results for functional balance measures than for postural control. However, heterogeneity was generally high, and the low number of studies included in this synthesis compel us to interpret these findings with caution (Figure 4).

Pooled data from two clinical trials [9,47] point to the superiority of VR and FB rehabilitation for gait speed improvements in pwDPN, although statistical significance was not reached. Moreover, the low number of studies included in this analysis, the different assessment tools used (TUG and 10 mWT) [9,47], and the high heterogeneity observed prevent us from considering these findings as conclusive (Figure 5).

Finally, publication bias could be assessed only by funnel plot asymmetry inspection (Figure 6 and Figure 7), as Begg’s or Egger’s statistical tests are recommended when at least 10 studies are included in each meta-analysis [63].

## 4. Discussion

This systematic review and meta-analysis aimed to summarize the current evidence regarding the effects of VR and/or FB on balance and gait parameters in pwDPN. Overall, interventions appeared to improve balance and gait outcomes. However, when compared with other interventions or no intervention, statistically significant differences have not been consistently observed.

Even though various tools have been used in clinical practice for diagnosing and assessing the severity of DPN, we highlighted the lack of standardized criteria in the assessment and categorization of sensory deficiencies, gait impairments, and balance disorders through the 10 included studies. There is a crucial need to establish standardized evaluation that specifically categorizes the phases and progression of balance control in pwDPN using validated and objective approaches [64]. 

Regarding the quality and duration of interventions, Grewal et al. [59] observed improved balance and coordination in both upper and lower limbs following a single session of approximately 15 min, showing promising outcomes through VR coupled with visual and auditory FB, even in the short term. Similarly, Chandrashekhar et al. [55] reported comparable results from a haptic FB intervention lasting only 10 min once a day. While these findings show promise, there is a need for studies with larger samples and controlled randomized design to compare treatment efficacy compared with standard care with varying dosages. Additionally, the absence of quality follow-ups impedes the assessment of sustainability of these effects over time, which constitutes an essential consideration for pwDPN, given that DPN is a chronic condition that can be managed but will worsen due to metabolic and ischemic deficit over the years. 

Heterogeneity was identified in the outcomes, interventions, and methodology measures among the selected studies meeting the inclusion criteria. A more specific analysis by study outcome measures is detailed below:

### 4.1. Balance

The age of the patients in the included studies ranged from 49 to 81 years. Stolarczyk et al. [56] reported an improvement in static balance among older patients (73 years old), suggesting encouraging outcomes in older pwDPN, which could be promoted in this population [15]. While it appears crucial to address proprioception through methods like unstable surfaces and conventional methods such as sensory stimulation of the sole of the foot and sensory FB [65], auditory and visual FB could play a pivotal role in improving static balance. This becomes particularly important as compensatory strategies involving vision and vestibular systems may be insufficient in pwDPN [66].

Grewal et al. [17] created an interesting paradigm with proactive balance training involving obstacle-overcoming exercises. However, the authors did not report statistically significant change in the risk of falls. Stolarczyk et al. [56] used a specifically proactive test with an unstable platform as a measure of risk of fall. Additionally, Salsabili et al. [60] demonstrated that training with dynamometric platforms helps in reintegrating sensory losses at a central level. Analyzing frequency components of CoM oscillation provides information on balance control: low frequencies correspond to visual components, medium frequencies are related to somatosensory activity, and high frequencies (above 1 Hz) indicate dysfunction at a central level due to altered afferent information. Similarly, Mohieldin et al. [58] investigated various components of balance (static balance, visual and vestibular components) using the SMART Balance Master platform. The authors concluded that training with visual FB on this platform not only improved balance at a functional level, but also improved its vestibular component. In this study [58], FB sessions were fewer in comparison with conventional therapy sessions, supporting the hypothesis that FB provided significant benefits in treatment of pwDPN. This differentiation between varied components of balance and the involvement of FB in recovery at the central and somatosensory levels represent a promising avenue of study and assessment tool for pwDPN. Regarding visual components, nearly all the studies excluded retinopathies or glaucoma from their sample, potentially biasing the results since these conditions are prevalent comorbidities in pwDPN [22]. Consequently, open eye conditions tend to show better outcomes than closed eye conditions, suggesting the need to explore auditory and haptic FB. In addition, the pain experienced by pwDPN could be a confounding factor when evaluating postural control [67] after a rehabilitation program. In this way, VR could potentially offer a dual effect by alleviating pain while improving postural control [68], emphasizing the importance of documenting pain in future studies.

### 4.2. Gait

Short-term loss of plantar sole sensation affects lower-limb kinematics and gait dynamics, especially in the initial stance phase, leading to altered muscle activation patterns during locomotion [4]. Thereby, the reduced plantar-afferent FB in pwDPN not only affects balance, but also influences kinematic aspects that could be improved through VR interventions.

In older pwDNP (72 years), Taveggia et al. [10] proposed a multimodal intervention and demonstrated statistically significant improvement in the intervention group compared to the control group for gait resistance, although no significant change in walking speed was noted (this study presented high-quality methodology). These findings suggest that a VR intervention, mainly focusing on gait, might facilitate muscular strengthening, thereby enhancing resistance. In a recent study, Venkataram et al. [69] reported consistent results with improved functional status following short-term strength and balance training for pwDNP, without any change in quality of life. However, Haimanot and Melaku [70] showed gait improvement using multimodal training encompassing strengthening, flexibility, stretching, and balance programs for pwDPN. Further research is needed, particularly to ascertain the response of intervention dosage and compare it with conventional therapy to establish a high level of evidence. The inclusion of FB intervention should be considered to optimize the impact of VR interventions.

Some studies were excluded [71,72,73,74,75] due to their lack of specific intervention during gait and/or objective measurement, as they focused on plantar pressures as a means for preventing ulcers. These studies intended to change a patient’s gait pattern by providing a larger area of support, thereby redistributing plantar pressures to prevent ulcers. However, none of them were dedicated to rehabilitation or assessment of the new gait pattern. It would be crucial to measure gait parameters and ensure quality intervention for patients, not only to prevent ulcers but also to manage new gait patterns that might potentially increase the risk of falls, which is already high in pwDPN. This avenue of research is compelling as it directly impacts two primary comorbidities that contribute to decreased quality of life and increased pain in pwDPN: risk of falls and ulcers. 

### 4.3. Risk of Falls

Two studies objectively evaluated the risk of falling [56,57], and showed statistically significant reduction in the risk of falls following VR intervention. These studies presented the most robust outcomes as they were rated the highest in terms of methodological quality among the studies included in this review. However, while Grewal et al. [17] and Carroll & Galles [27] did not reported specific risk of falls measurements, the studies showed statistically significant improvement in the SMF-12 mental score and the ABC test, respectively. While these findings are promising as means of enhancing the coping mechanisms for fear of falling among pwDPN, caution should be exercised in interpreting them, especially with Carroll & Galles [27], which presents a single case and has lower methodological quality. Visual component disorders in pwDPN are once again emphasized, as they are associated with an increased risk of falls and fear of falling. Although the predominant type of FB used in the studies was visual, exploring auditory and haptic FB could be an important avenue for evaluating fear and fall risk. 

Given the importance of muscle spindles in the proprioception of pwDPN [7], improvements in proprioception could be potentiated with auditory and visual FB. In addition, the close relationship between muscle spindles and sensitivity to plantar sole sensitivity [7] offers an interesting way of intervention to prevent complications such as ulcers and thereby reduce the risk of falls [8,26]. Exploring VR rehabilitation incorporating haptic FB alongside auditory/visual FB could be pivotal.

### 4.4. Study Limitations and Future Research Lines

This systematic review and meta-analysis have evident limitations in conclusively establishing the effectiveness of VR due to the limited number of studies with high methodological quality. However, the synthesized and in-depth analysis presented can serve as a roadmap for future randomized controlled studies. Future research directions could focus on conducting studies with larger sample sizes, more standardized evaluations, and stratification based on deficits (e.g., patients with or without retinopathy) according to the severity levels of DPN.

## 5. Conclusions

Pooled results indicate a positive trend in favor of VR and FB rehabilitation in pwDPN, yet significant differences have not been reached for balance or gait speed improvements. Auditory and visual FB could play a crucial role in improving static balance, especially as compensation strategies related to vision and vestibular function in pwDPN. Dynamometric platforms, particularly those incorporating haptic FB, have shown good outcomes, even among older patients, by improving walking resistance and reducing the risk of falls. The addition of visual FB appears more beneficial in maintaining balance (mainly static and proactive), as the visual component can be managed to adapt balance strategy in open-eye conditions. Future studies should consider stratifying groups based on the presence or absence of diabetic retinopathy to gain better insights into these effects. Additionally, more studies focused on gait, including endurance improvement, are warranted. Concerning dosage, promising results have been observed with very short interventions and a limited number of sessions. While there is insufficient data to definitively state whether VR or FB interventions are more effective than conventional therapy, these systems offer a broader evaluative spectrum, enabling differentiation, classification and more specific action on various gait and balance parameters.

## Figures and Tables

**Figure 1 healthcare-11-03037-f001:**
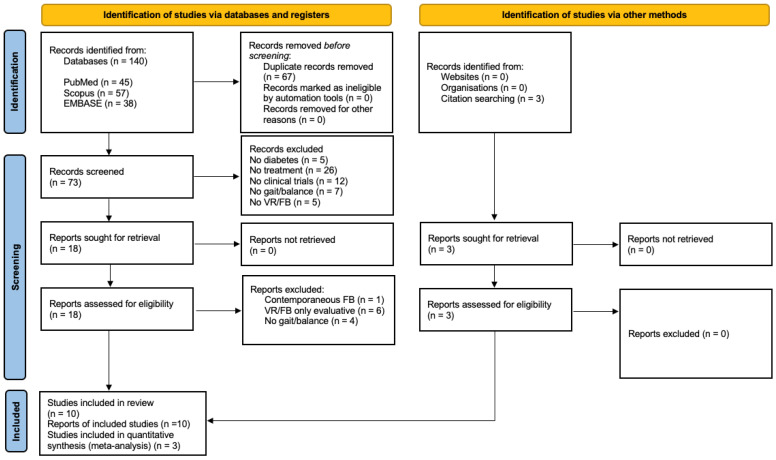
Flowchart of the different phases of the systematic review [49].

**Figure 2 healthcare-11-03037-f002:**
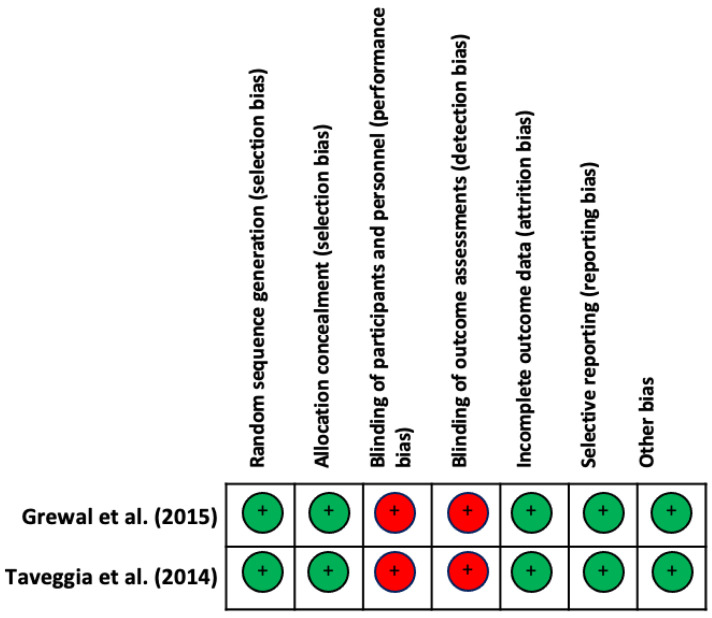
Risk of bias [54] of the studies included in the systematic review [10,17]. Green: Low risk of bias; Yellow: Unclear risk of bias; Red: High risk of bias.

**Figure 3 healthcare-11-03037-f003:**
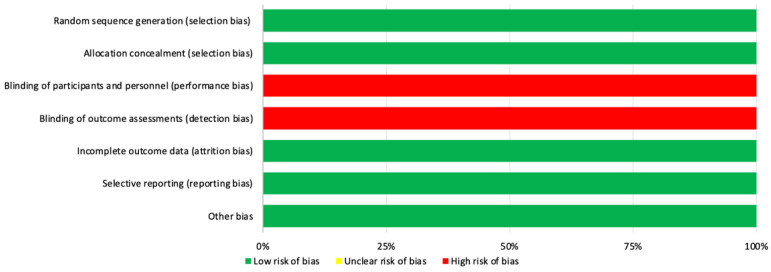
Overall risk of bias [54]. The results are presented by percentages.

**Figure 4 healthcare-11-03037-f004:**
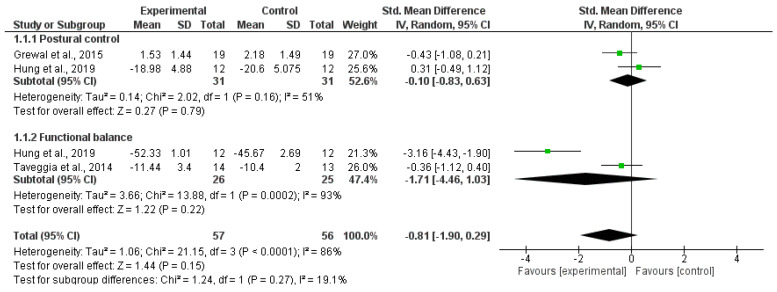
Forest plot of the comparison of the difference between the subgroups in postural control and functional balance [10,17,57].

**Figure 5 healthcare-11-03037-f005:**
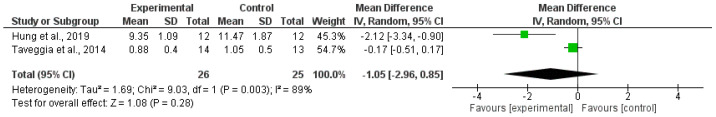
Forest plot of the comparison of the difference between the subgroups in gait speed [10,57].

**Figure 6 healthcare-11-03037-f006:**
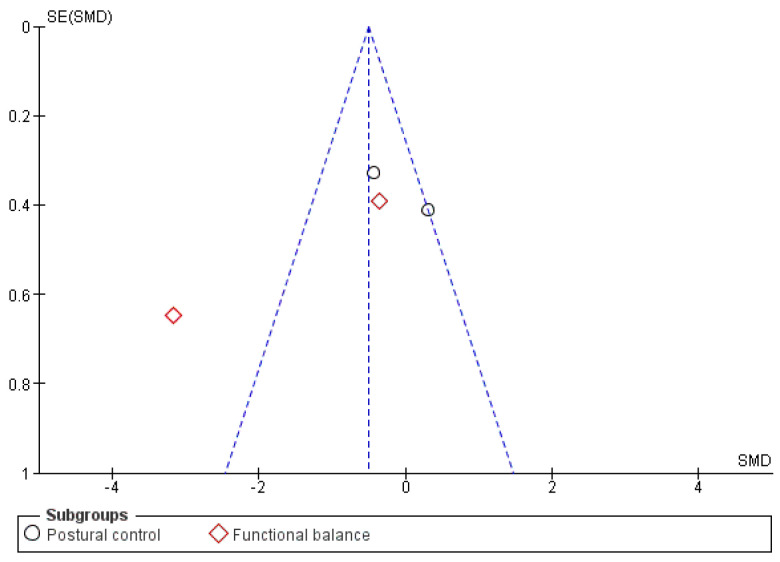
Funnel plot for balance. Where both slanting dotted lines represent the expected 95% CI; vertical dotted line referring to the position of no effect; points represent individual studies.

**Figure 7 healthcare-11-03037-f007:**
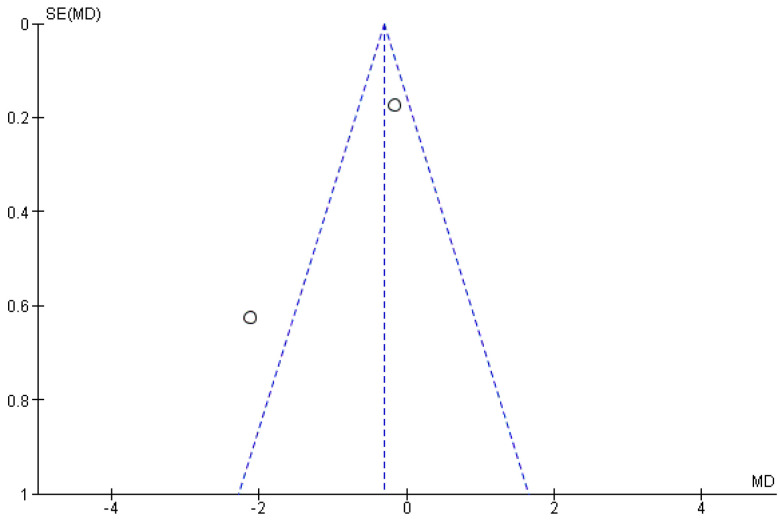
Funnel plot for gait speed. Where both slanting dotted lines represent the expected 95% CI; vertical dotted line referring to the position of no effect; points represent individual studies.

**Table 1 healthcare-11-03037-t001:** Search strategy for the different databases.

Databases	Search
Medline/PubMed Scopus	(**“virtual reality”** OR **feedback** OR **biofeedback** OR **wii** OR **game** OR **“augmented reality”** OR **kinect** OR **videogam*** OR **“virtual training”** OR **“immersive training”**) AND **(“diabetic neuropath*”** OR **“diabetic polineuropath*”**) AND (**gait** OR **balance** OR **walking** OR **“risk of fall”**)
EMBASE	(**‘virtual reality’**/exp OR **‘virtual reality’** OR **‘virtual reality exposure therapy’**/exp OR **‘virtual reality exposure therapy’** OR **‘video game’**/exp OR **‘video game’** OR **‘feedback system’**/exp OR **‘feedback system’**) AND (**‘diabetic neuropathy’**/exp OR **‘diabetic neuropathy’**) AND (**‘gait’**/exp OR **gait** OR **‘balance’**/exp OR **balance** OR **‘falls’**/exp OR **falls**)

**Table 2 healthcare-11-03037-t002:** Score obtained on the PEDro scale [51] of each of the selected studies.

Study	Total	1	2	3	4	5	6	7	8	9	10	11
Grewal et al., 2015 [17]	7/10	-	1	1	1	0	0	0	1	1	1	1
Taveggia et al., 2014 [10]	7/10	-	1	0	1	0	0	1	1	1	1	1

**Table 3 healthcare-11-03037-t003:** Score obtained on the CMSQ scale [53] of each of the selected studies.

Study of Methodological Quality (Checklist for Measuring Quality), and Grade of Recommendation (Guidelines of the Oxford Centre for Evidence-Based Medicine).										
	Chadrashekhar et al., 2021 [55]	Stolarczyk et al., 2021 [56]	Hung et al., 2019 [57]	Carroll & Galles, 2018 [27]	Grewal et al., 2015 [17]	Mohieldin et al., 2014 [58]	Taveggia et al., 2014 [10]	Grewal et al., 2013 [59]	Salsabili et al., 2013 [60]	Salsabili et al., 2011 [9]
**Checklist items for Measuring Quality**										
	**1. Study quality**									
Hypothesis/aim	1	1	1	0	1	1	1	1	1	1
Outcomes	1	1	1	1	1	1	1	1	1	1
Eligibility criteria	1	1	1	0	1	1	1	1	1	1
Interventions	1	1	1	1	1	1	1	1	1	1
Confounders	0	0	0	0	0	0	0	0	0	0
Findings	1	1	1	0	1	1	1	1	1	1
Random variability	0	0	0	0	0	0	0	0	0	0
Adverse events	0	0	1	0	0	0	1	1	0	0
Lost to follow-up	0	1	1	0	0	0	1	1	0	0
Probability values	0	1	1	0	0	0	1	1	0	0
	**2. External validity**									
Source population	1	1	1	0	1	1	1	1	1	1
Illustrative sample	0	0	0	0	0	0	0	0	0	0
Illustrative treatment	1	1	1	1	1	1	1	1	1	1
	**3. Internal validity (study bias)**									
Blinding of subjects	1	0	0	0	0	0	1	0	0	0
Blinding	0	0	0	0	0	0	1 0	0	0	0
“Data extraction”	0	0	0	0	0	0	0	0	0	0
Follow-up adjustments	0	0	0	0	0	0	0	0	0	0
Statistical tests	1	1	1	1	1	1	1	1	1	1
Compliance	1	1	1	1	1	1	1	1	1	1
Outcomes	1	1	1	0	1	1	1	1	1	1
	**4. Internal validity (confounding and selection bias)**						1			
Source of patients	1	1	1	0	1	1	1	1	1	1
Recruitment period	1	1	1	0	1	1	1	1	1	1
Randomization	1	0	1	0	1	1	1	0	0	0
Concealment	0	0	1	0	1	1	1	0	0	0
Analysis	0	0	0	0	0	0	0	0	0	0
Loss to follow-up	0	1	1	0	0	0	0	1	0	0
	**5. Power**									
Effect	2	5	2	0	5	5	5	2	2	2
**Total score**	16	20	19	5	19	19	22	18	14	14
Percentage (%)	50	62	59	16	59	59	69	56	44	44

**Table 4 healthcare-11-03037-t004:** Synthesis of results.

Authors (Year) Design	Sample	Age (Average, Standard Dev.)	Intervention	N Ses	Performance of Measurement	VR	FB	Results
Chandrashekhar et al., (2021) [55] PS	N = 23	66.6 ± 9.9	Vibration wearables (Myovolt)	10 min 3 ses/week 4 weeks	Static and dynamic Balance (BSS, TUG) Proactive Balance (TUG-Cognitive)Pain (BPI-DPN)	-	Haptic	↑ TUG y TUG-Cognitive
Stolarczyk et al., (2021) [56] N-RCT	N = 77	73.22 ± 7.57	BSSys	30 min 5 ses/week 12 weeks	Static and Dynamic Balance (2 BSSys tests EO and EC: general data, AP, and ML) Risk of Fall (BSSys) Motor coordination (BSSys)	-	Visual and Haptic	↑ static balance EO and EC, all parameters ↓ risk of fall All parameters were significantly worse in EG (DPN patients) than CG (healthy participants)
Hung et al., (2019) [57] CCT	N = 24 A = 12 B = 12	49.9 ± 12	Interactive game + mat (XavixPORT)	30 min 3 ses/week 6 weeks	Static and Dynamic Balance (TUG, BSS) Risk of Fall (MFES) Monopodial Balance (UST)	Semi- immersive	Visual and auditive	↑ BSS, MFES y TUG. = UST
Carroll & Galles (2018) [27] CC	N = 1	60	3 interactive games (OGRE)	45 min 6 ses	Balance (TUG, 2° session) Security (ABC)	Semi-immersive	Visual and auditive	↑ TUG ↑ ABC Plus positive verbal feedback
Grewal et al., (2015) [17] RCT	N = 35 CG = 20 EG = 19	63.7 ± 8.2	CG: standard care EG: interactive game (MATLAB), 5 sensors (LegSys) hips and lower limbs.	45 min 2 ses/week 4 weeks	Risk of Fall (FES-I) Quality of life (SF-12) Balance (double stance, CoM AP sway, LM sway and total sway) Daily physical activity (PAMSys)	Non- immersive	Visual and auditive	↓ CoM total sway and ML sway EO ↓ sway hips–ankles ↑ mental punctuation SF-12
Mohieldin et al., (2014) [58] N-RCT	N = 87 CG = 30 EG = 57	49.9 ± 12	CG: standard care EG: Dynamic posturography (SMART Balance Master)	CG: 3 ses/week, 12 weeks EG: 2 ses/week, 12 weeks	Static balance: visual and vestibular proportion EO and EC (SMART Balance Master)	-	Visual	↓ compost balance and SMTS relation DPN light and CG. ↓ compost balance and SMTS relation DPN severe and CG. ↑ pre-post SMTS relation DPN light.
Taveggia et al., (2014) [10] RCT	N = 27 CG = 14 EG = 13	72 ± 9	CG: standar care EG: Biodex protocol	60 min 5 ses/week 4 weeks	Gait (6MWT, 10MWT and Tinetti) Function (FIM and Tinetti) Balance (Tinetti)	-	Visual and haptic.	↑ walking resistance EG ↑ functionality (FIM) both groups
Grewal et al., (2013) [59] PCS	N = 29	57 ± 10	Interactive gam (MATLAB) 6 inertial sensors (BalanSens) ankles and wrist.	10–15 min 1 ses	Balance (CoM sway = BalanSens) Coordination between upper and lower limbs AP and ML (RCI)	Non- immersive	Visual and auditive	↓ CoM hips-ankles sway, EC. ↓ CoM Sway, EO. ↑ coordination AP EO.
Salsabili et al., (2013) [60] T-SS	N = 19	56 ± 8.96	BSSys	30 min 10 ses/week 3 weeks	Static Balance (torque platform) DPN severity (Valk score)	-	Visual	= Valk Score: no ↓ DPN severity. ↓ CoM sway ML low-medium frequencies OE.
Salsabili et al., (2011) [9] T-SS	N = 19	56 ± 8.96	BSSys	30 min 10 ses/week 3 weeks	DPN severity (Valk score) Balance (CoM fluctuation = BSSys, 3 trials EO and 3 EC)	-	Visual	= Valk Score: no ↓ DPN severity. ↑ stability total, AP, and ML EO. ↑ stability total and AP EC

(ABC) Activities-Specific Balance Confidence Scale; (AP) Anteroposterior; (BSS) Berg Scale Score; BSSys Biodex Stability System; (BPI-DPN) Brief Pain Inventory for patients with painful Diabetic Peripheral Neuropathy; (CC) Clinical Case; (CCT) Crossover Clinical Trial; (CG) Control Group; (CoM) Center of Mass; (DPN) Diabetic Peripheral Neuropathy (EC) Eyes Closed; EO (Eyes Opened); (EG) Experimental Group; (FES-I) Falls Efficacy Scale; (FIM) Functional Independence Measure; (MFES) Modified Falls Efficacy Scale; (ML) Medial-lateral; (min) minutes; (N-RCT) Non-Randomized Controlled Trial; (PCS) Prospective Cohort Study; (PS) Pilot Study; (RCI) Reciprocal Compensatory Index; (RCT) Randomized Controlled Trial; (SF-12) Short-form health survey; (Ses) session; (SMTS) somatosensorial; (T-SS) Time-Series Study; (TUG) Timed Up and Go; (6MWT) 6-Minute Walking Test; (10 MWT) 10-Meter Walking Test.

## Data Availability

Data are contained within the Appendix A.

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
