# Peer review of "Effectiveness of Virtual Reality and Feedback to Improve Gait and Balance in Patients with Diabetic Peripheral Neuropathies: Systematic Review and Meta-Analysis"

_healthcare, 2023, doi:10.3390/healthcare11233037_

Round 1

Reviewer 1 Report

Comments and Suggestions for Authors

Dear Authors,
thank you very much for your paper. The relevance of the topic cannot be neglected with the demographic change and the increase of diabetic diseases. Your article gives a good overview and clearly shows the decoupling between theory and practice. Because the diagnostics and therapy are part of the daily work since many years, but obviously there are still too few standardized data recordings to be able to study them better.

In the following, I would like to give you a few hints on how to enhance the article.

Introduction:

- Line 52 f.: heightended might be misleading. "Increased" would probably be clearer.

- same paragraph: lt might be interesting to explain in 2-3 sentences why balance testing often tests and contrasts eyes-up and eyes-down. It is certainly worth mentioning our dominant dependence on the visual sense, but it is the result under closed eyes that is important for these patients. Because then we reduce the input to the other systems that we want to train in particular. Perhaps you also have a comparative value of what the ratio of the result with eyes open and eyes closed should be in terms of Sway-Path-Length or Area-of-Elipse.

- Line 59: Please clarify how therapy affects morbidity.

Materials and Methods:

- Very good and transparent presentation.

- 2.1. Search Strategy: Clarify why you chose different search strategies in dependence on the databases.

- Line 146: the sentence in misleading, since you did your search with several databases but you're only mentioning PubMed.

Results:

- Please check the placement of the subheadlines in table 3.

- Typo: Visual and Haptic vs. Visual and auditive (Capitalization)

Discussion:

- Line 288: Didn't you want to say, that CoM increased?
- Line 309: At the latest here, as a reader of your article, you would want to learn a little bit about the background.

- Line 344: Risk of falls

In general a good article with good methods. In my opinion, in the respective sections of the discussion (balance, gait, risk of falls) 2-3 sentences should be said about physiological context. You are strongly oriented to the methodological quality of the included studies, also their results are named, but the gap between method and pathology could be better explained functionally. In particular, how the disease affects exteroception (Vater-Paccini corpuscles, etc.) and proprioceptors (muscle spindles, Golgi tendon apparatuses).

Comments on the Quality of English Language

Dear authors,
I recommend to check the document again carefully for typos, double spaces and grammatical errors. I am not a native speaker myself, but I would like to take the opportunity to point out a few lines where there is a typo or an error in expression. Lines: 27, 148, 202, 218, 230, 234, 239, 270, 288, 292, 294, 300, 303, 325, 350, 384.

Author Response

REVIEWER 1

1.0. Dear Authors,
thank you very much for your paper. The relevance of the topic cannot be neglected with the demographic change and the increase of diabetic diseases. Your article gives a good overview and clearly shows the decoupling between theory and practice. Because the diagnostics and therapy are part of the daily work since many years, but obviously there are still too few standardized data recordings to be able to study them better.

Thank you very much for your comments and ratings.

In the following, I would like to give you a few hints on how to enhance the article.

- Introduction:

1.1. Line 52 f.: heightended might be misleading. "Increased" would probably be clearer.

Done.

1.2. same paragraph: lt might be interesting to explain in 2-3 sentences why balance testing often tests and contrasts eyes-up and eyes-down. It is certainly worth mentioning our dominant dependence on the visual sense, but it is the result under closed eyes that is important for these patients. Because then we reduce the input to the other systems that we want to train in particular. Perhaps you also have a comparative value of what the ratio of the result with eyes open and eyes closed should be in terms of Sway-Path-Length or Area-of-Elipse.

Lines 48-68. Phrases with updated references have been included that explain the questions requested at this point.

1.3. Line 59: Please clarify how therapy affects morbidity.

The phrase "(by preventing foot deformities and improving plantar load distribution that are related to areas of high pressure during walking that increase the risk of ulceration) [28,29]" has been included to clarify this issue.

- Materials and Methods:

1.4. Very good and transparent presentation.

Thank you very much for your appreciations.

1.5.   2.1. Search Strategy: Clarify why you chose different search strategies in dependence on the databases.

The search strategy has been unified, including the EMBASE database with its specific search with the terms recommended in this database.

1.6. Line 146: the sentence in misleading, since you did your search with several databases but you're only mentioning PubMed.

Thank you for your question. In this line of the manuscript we describe how we performed a pre-search in PubMed to locate the most appropriate descriptors and define a definitive search strategy. The terms presearch and definitive search have been included; we believe that this clarification will avoid confusion for readers.

- Results:

1.7. Please check the placement of the subheadlines in table 3.

Done.

1.8. Typo: Visual and Haptic vs. Visual and auditive (Capitalization)

I apologize, I don't understand the question.

- Discussion:

1.9. Line 288: Didn't you want to say, that CoM increased?

In the studies there was a decrease in CoM, which meant greater post-intervention stability.

1.10. Line 309: At the latest here, as a reader of your article, you would want to learn a little bit about the background.

The phrase “Globally, DR is also a prevalent complication among patients with newly diagnosed type 2 DM [7] that, together with dysfunction of the vestibular system, causes balance deficits and an increased risk of falls in people with DPN (pwDPN) [8]” has been included at the end of the first paragraph to clarify this aspect in the Introduction section, and the reference has been updated.

1.11. Line 344: Risk of falls

The term has been changed.

1.12. In general, a good article with good methods. In my opinion, in the respective sections of the discussion (balance, gait, risk of falls) 2-3 sentences should be said about physiological context. You are strongly oriented to the methodological quality of the included studies, also their results are named, but the gap between method and pathology could be better explained functionally. In particular, how the disease affects exteroception (Vater-Paccini corpuscles, etc.) and proprioceptors (muscle spindles, Golgi tendon apparatuses).

Thanks for the appreciation. Clarifications have been included in the final part of the first paragraph of the Introduction section, as well as in the Discussion sections.

Comments on the Quality of English Language

Dear authors,
I recommend to check the document again carefully for typos, double spaces and grammatical errors. I am not a native speaker myself, but I would like to take the opportunity to point out a few lines where there is a typo or an error in expression. Lines: 27, 148, 202, 218, 230, 234, 239, 270, 288, 292, 294, 300, 303, 325, 350, 384.

Thank you very much for the appreciation. These errors have been corrected and/or the phrases that were not well expressed in English have been clarified.

Reviewer 2 Report

Comments and Suggestions for Authors

Dear authors,

Thank you for the opportunity to review this work. This systematic literature review is interesting, and I think it is relevant to the scientific community and a valuable addition to Virtual Reality applications in Rehabilitation. The VR literature needs more research to establish evidence-based practice. However, I have noted some areas of improvement as follows:

Abstract

In line 27, the authors mentioned this: “Therapy through RV …” I think the authors mean VR but not RV. Please fix this typo.

Introduction

In line 88, the authors cited this statement: “No specific systematic review on the use of VR has been carried out”; however, this is not true. One recent systematic review and meta-analysis examined VR applications specifically on static and dynamic balance. Your systematic review is a precious addition to the previous work, and you should add this systematic review to your discussion in the introduction to widen the scope of VR applications and support your main argument:

(Elaraby, A. E. R., Shahien, M., Jahan, A. M., Etoom, M., & Bekhet, A. H. (2023). The Efficacy of Virtual Reality Training in the Rehabilitation of Orthopedic Ankle Injuries: A Systematic Review and Meta-analysis. Advances in Rehabilitation Science and Practice, 12, 11795727231151636. https://doi.org/10.1177/11795727231151636)

Methods

Line 98: I wonder why the authors excluded Embase from their search.

Line 116, The Assessment of the Methodological Quality and Risk of Bias section, is wordy and hard to follow with some language mistakes. I suggest re-writing this section.

Line 145, it is unclear who selected articles during the Selection Process and Data Extraction. You should provide the number of researchers who did the screening, who validated the screening, and who resolved any conflicts between the screeners.

In lines 148-149, there are Chinese words that I don’t understand. Please clarify in English what these words mean.

Results

In the abstract, you mentioned “153 studies identified and 8 were included”, but in the results (lines 159-160), you cited 186 studies were identified, and 10 were included.” so which one is the correct number? Please double-check and revise.

Did the authors exclude any articles due to the poor quality? If yes, please add a description to this section.

It is confusing why the authors used two quality assessment tools (the PEDro scale & the CMSQ). Please provide a justification because these two tools can be used for clinical trials.

The results should provide a qualitative description of the study design for all the included studies. Please address.

In Table 4, they cited this “ Age (Average, typical deviation).” what do you mean by typical deviation? Did you mean standard deviation? Please clarify.  

Line 220, (3.3.2. Training sessions). This section is extremely difficult to follow. I think putting all these training protocols in one basket is confusing. I suggest discussing similar ideas in one paragraph and then going to the next paragraph with new ideas. There are several language errors in this paragraph as well. For example, what do you mean by these question marks at the end of this sentence fragment: “First with parallel feet, and then changing the advanced one??”  

The results miss a significant paragraph about the effectiveness of VR applications on the outcomes of interest. It would help if you wrote up a paragraph before the discussion.

Discussion

You mentioned in lines 90-91 of the introduction that the aim of your study was “The present study proposes a review and synthesis of the existing evidence to date on the effects 91 of both VR and FB in improvement of gait and balance in pwPDN.”

I think your results and discussion do not achieve this aim.

Your aim should be in the first line of your discussion.

I don’t think you were interested in the diagnosis and assessment of the severity of DPN! Why did you spend too much time discussing these things??!

Line 270, Gurtej et al. [49] is wrongly placed. In the references list, reference# 49 is Grewal et al., not Gurtej et al.

And by the way, Grewal et al. examined the effects of visual feedback on balance by assessing ankle joints. Most participants in their study were older adults (mean age = 57 years), raising generalizability questions.

Line 272-273, this conclusion is vague and should be carefully re-worded: “which could indicate that training through VR with visual and auditory FB is a powerful tool even in the short term.”

Line 302, which study you were referring to in this sentence: “In this study, FB sessions were fewer”?

In line 360 (4.4. Study Limitations), you should mention all your study limitations and make it clear that several methodological limitations may bias your results (please see my previous comments).

Also, how did you come up with this: “This review has clear limitations due to the scarcity of studies with a high level of evidence.”

Please discuss all of the following questions in your limitations section:

-          Why didn’t you perform a meta-analysis?

-          Do you have a possibility of missing any relevant articles?

-          Don’t you think searching literature up to Nov 2022 is not a limitation?

-          Don’t you think your search strategy might be biased?

Since this is a systematic review, you must be clear and transparent in every bias you might have.

Line 363, what do you mean by representative population in this statement: “Future lines of research could focus their efforts on implementing studies with representative populations.”

Conclusion

The conclusion is quite long. I suggest reducing this paragraph to data-driven and straightforward conclusions.

All the best,

Comments on the Quality of English Language

The English language of this manuscript needs an extensive review. 

Author Response

REVIEWER 2

Dear authors,

2.0. Thank you for the opportunity to review this work. This systematic literature review is interesting, and I think it is relevant to the scientific community and a valuable addition to Virtual Reality applications in Rehabilitation. The VR literature needs more research to establish evidence-based practice. However, I have noted some areas of improvement as follows:

Thank you very much for your comments and ratings.

Abstract

2.1. In line 27, the authors mentioned this: “Therapy through RV …” I think the authors mean VR but not RV. Please fix this typo.

We apologize for the mistake. Thank you very much for your appreciation. It has been modified.

Introduction

2.2. In line 88, the authors cited this statement: “No specific systematic review on the use of VR has been carried out”; however, this is not true. One recent systematic review and meta-analysis examined VR applications specifically on static and dynamic balance. Your systematic review is a precious addition to the previous work, and you should add this systematic review to your discussion in the introduction to widen the scope of VR applications and support your main argument:

(Elaraby, A. E. R., Shahien, M., Jahan, A. M., Etoom, M., & Bekhet, A. H. (2023). The Efficacy of Virtual Reality Training in the Rehabilitation of Orthopedic Ankle Injuries: A Systematic Review and Meta-analysis. Advances in Rehabilitation Science and Practice12, 11795727231151636. https://doi.org/10.1177/11795727231151636)

Thanks for the appreciation. “No specific systematic review on the use of VR in pwDPN” has been added to clarify this specificity and the proposed review has been included with the text “[38], as well as in the rehabilitation of orthopedic ankle injuries”. In addition, another specific review of non-pharmacological treatment in pwDPN through offloading devices has been added (Horstink et al. 2021) to further contextualize the evidence.

Methods

2.3. Line 98: I wonder why the authors excluded Embase from their search.

The EMBASE database has been included in the search.

2.4. Line 116, The Assessment of the Methodological Quality and Risk of Bias section, is wordy and hard to follow with some language mistakes. I suggest re-writing this section.

Thanks for your appreciation. This section has been rewritten and shortened.

2.5. Line 145, it is unclear who selected articles during the Selection Process and Data Extraction. You should provide the number of researchers who did the screening, who validated the screening, and who resolved any conflicts between the screeners.

Thanks for your appreciation. The initials of the names of the authors who carried out the searches and the consensus have been included: “Two authors (L.A.-E. and L.G.-C.) carried out the screening process, and an additional reviewer (C.L.-M.) was considered for consensus when needed”.

2.6. In lines 148-149, there are Chinese words that I don’t understand. Please clarify in English what these words mean.

We apologize for the confusion with the language. Presearch and definitive search have been included to better clarify the steps followed and “descriptors” have been replaced by “search terms”.

Results

2.7. In the abstract, you mentioned “153 studies identified and 8 were included”, but in the results (lines 159-160), you cited 186 studies were identified, and 10 were included.” so which one is the correct number? Please double-check and revise.

Thanks for your appreciation. We apologize for the error in the Abstract, the correct data are those included in the manuscript, flow chart and supplementary list.

2.7. Did the authors exclude any articles due to the poor quality? If yes, please add a description to this section.

Given the scarcity of articles, in an effort to thoroughly review the existing evidence, we have not only included high-quality randomized controlled studies. However, we have referred to the strength of the results based on the methodological quality of the studies in the Discussion section.

2.8. It is confusing why the authors used two quality assessment tools (the PEDro scale & the CMSQ). Please provide a justification because these two tools can be used for clinical trials.

It has been justified: “The evaluation of methodological quality was carried out through the Physiotherapy Evidence Database (PEDro) scale [51] in the studies in which it was possible to implement this scale (RCTs), and the Checklist for Measuring Study Quality (CMSQ) [40] for all the studies included (to have a broader vision of all types of studies such as case series, etc.)”.

2.9. The results should provide a qualitative description of the study design for all the included studies. Please address.

It has been included.

2.10. In Table 4, they cited this “Age (Average, typical deviation).” what do you mean by typical deviation? Did you mean standard deviation? Please clarify.  

Thank you very much for your appreciation. We apologize for the mistake. It has been corrected.

2.11. Line 220, (3.3.2. Training sessions). This section is extremely difficult to follow. I think putting all these training protocols in one basket is confusing. I suggest discussing similar ideas in one paragraph and then going to the next paragraph with new ideas. There are several language errors in this paragraph as well. For example, what do you mean by these question marks at the end of this sentence fragment: “First with parallel feet, and then changing the advanced one??”  

Thanks for your feedback. We agree with your comments and, following your suggestions, we have divided the articles by paragraphs, paying attention to the expression.

2.12. The results miss a significant paragraph about the effectiveness of VR applications on the outcomes of interest. It would help if you wrote up a paragraph before the discussion.

Has been included:” This systematic review and meta-analysis aimed to summarize the evidence to date on the effects of VR and/or FB for balance and gait parameters in pwDPN. In general, both interventions seem to improve these outcomes, but when data is compared with other interventions/no intervention, statistically significant differences have not been observed”.

Discussion

2.13. You mentioned in lines 90-91 of the introduction that the aim of your study was “The present study proposes a review and synthesis of the existing evidence to date on the effects 91 of both VR and FB in improvement of gait and balance in pwPDN.”

I think your results and discussion do not achieve this aim.

We have updated the search and included important methodological aspects such as the meta-analysis and we hope, now, that the proposed objective has been achieved.

2.14. Your aim should be in the first line of your discussion.

The objective has been included in the first line of the discussion.

2.15. I don’t think you were interested in the diagnosis and assessment of the severity of DPN! Why did you spend too much time discussing these things??!

Many virtual reality systems have both diagnostic and interventional use. It is important to delve deeper into this topic, since consensus on the evaluation criteria to well stratify the sample with respect to the interventions is key for the implementation of future studies, as we point out in the conclusion.

2.16. Line 270, Gurtej et al. [49] is wrongly placed. In the references list, reference# 49 is Grewal et al., not Gurtej et al.

Thank you very much for your appreciation. It has been corrected.

2.17. And by the way, Grewal et al. examined the effects of visual feedback on balance by assessing ankle joints. Most participants in their study were older adults (mean age = 57 years), raising generalizability questions.

An overview of the age of the participants in the selected studies has been provided in results “participants and intervention characteristics” and discussion.

2.18. Line 272-273, this conclusion is vague and should be carefully re-worded: “which could indicate that training through VR with visual and auditory FB is a powerful tool even in the short term.”

It has been replaced by “, showing hopeful results with the use of VR with visual and auditory FB, even in the short term.”

2.19. Line 302, which study you were referring to in this sentence: “In this study, FB sessions were fewer”?

It was a continuation of the study by Mohieldin et al. This reference has been assigned again.

2.20. In line 360 (4.4. Study Limitations), you should mention all your study limitations and make it clear that several methodological limitations may bias your results (please see my previous comments).

Limitations have been addressed.

2.21. Also, how did you come up with this: “This review has clear limitations due to the scarcity of studies with a high level of evidence.”

Thanks for your appreciation. This phrase has been replaced by “This systematic review and meta-analysis has clear limitations in terms of emphatically concluding the effectiveness of VR due to the scarcity of studies with high methodological quality”, in an attempt to clarify the idea.

2.22. Please discuss all of the following questions in your limitations section:

2.22.a.  Why didn’t you perform a meta-analysis?

Following your suggestion, we have reviewed the articles again and thought it was possible to implement a meta-analysis and so we have included it.

2.23.b. Do you have a possibility of missing any relevant articles?

The strategy has been modified and the search updated in an effort to clarify whether there were new relevant studies that could have been included.

2.24.c. Don’t you think searching literature up to Nov 2022 is not a limitation?

The bibliographic search has been updated until November 2023.

2.25.d. Don’t you think your search strategy might be biased?

The search strategies have been modified to avoid bias.

2.23. Since this is a systematic review, you must be clear and transparent in every bias you might have.

The biases that this review presented have been eliminated with the modifications previously explained.

2.24. Line 363, what do you mean by representative population in this statement: “Future lines of research could focus their efforts on implementing studies with representative populations.”

It has been replaced by “Future lines of research could focus their efforts on implementing studies with a larger sample size, more homogenized evaluations, stratifying by deficits (for example, patients with or without retinopathy, etc.) according to degrees of severity of the DPN”.

Conclusion

2.25. The conclusion is quite long. I suggest reducing this paragraph to data-driven and straightforward conclusions.

The discussion has been rewritten and shortened.

 The English language of this manuscript needs an extensive review. 

A ProofReading has been carried out on the entire manuscript.

Round 2

Reviewer 1 Report

Comments and Suggestions for Authors

No futher comments needed.

Good work.

Reviewer 2 Report

Comments and Suggestions for Authors

Dear respected authors,

Thank you for your hard work. I do not have any comments to add. You did a great job in addressing our concerns. The revised version looks excellent and this kind of research is pretty much needed in the field of VR rehabilitation.

I wish you all the best,